# Udder Measurements and Their Relationship with Milk Yield in Pelibuey Ewes

**DOI:** 10.3390/ani10030518

**Published:** 2020-03-20

**Authors:** Darwin Arcos-Álvarez, Jorge Canul-Solís, Ricardo García-Herrera, Luis Sarmiento-Franco, Ángel Piñeiro-Vazquez, Fernando Casanova-Lugo, Luis O. Tedeschi, Manuel Gonzalez-Ronquillo, Alfonso Chay-Canul

**Affiliations:** 1División Académica de Ciencias Agropecuarias, Universidad Juárez Autónoma de Tabasco, 86280 Villahermosa, Mexico; darwin.arcos@itconkal.edu.mx (D.A.-Á.); ricardogarciaherrera@hotmail.com (R.G.-H.); 2Tecnológico Nacional de México/ Instituto Tecnológico de Conkal, 97345 Conkal, Mexico; pineiroiamc@gmail.com; 3Departamento de Posgrado e Investigación, Tecnológico Nacional de México/I. T. Tizimin, 97700 Tizimín, Mexico; jcanul31@gmail.com; 4Facultad de Medicina Veterinaria y Zootecnia, Universidad Autónoma de Yucatán, 97100 Mérida, Mexico; luis.sarmiento@correo.uady.mx; 5Instituto Tecnológico de la Zona Maya, Tecnológico Nacional de México, 77960 Othón P. Blanco, Mexico; fkzanov@gmail.com; 6Department of Animal Science, Texas A&M University, College Station, TX 77843-2471, USA; luis.tedeschi@tamu.edu; 7Departamento de Nutricion Animal, Facultad de Medicina Veterinaria y Zootecnia, Universidad Autonoma del Estado de Mexico, 50000 Toluca, Mexico; mrg@uaemex.mx

**Keywords:** udder measurements, milk yield, hair sheep, mathematical models

## Abstract

**Simple Summary:**

The Pelibuey sheep is considered the main maternal breed in the tropical production systems in Mexico. Nonetheless, there are few studies related to milk production and composition. The quantification of milk production in sheep is important because milk is the main source of nutrients for the growth, development and health of lambs. However, in hair sheep breeds, milking is very difficult due to the small size of their teats. Hence, it is important to evaluate indirect methods to estimate the milk yield in Pelibuey ewes to optimize the growth and to develop management strategies for the lambs.

**Abstract:**

The study aimed to evaluate the relationship between udder measurements and milk yield (MY) in dairy Pelibuey ewes. Udder measurements were taken twice a week for eight weeks before (initial) and after (final) milking, including udder depth (UD), udder circumference (UC), udder width (UW), teat length (TL) and teat diameter (TD) in 38 multiparous ewes. Additionally, udder volume (UV) and the difference (VDF) between initial UV (UVi) and final (UVf) was calculated as VDF = UVi − UVf. The MY varied from 0.10 kg/d to 1.04 kg/d, with a mean of 0.39 kg/d, ± 0.18 kg/d. Initial UC (UCi) ranged from 25.80 cm to 53.30 cm, and VDF varied from 1 cm^3^ to 2418 cm^3^. The TL and TD were not correlated with MY (*p* > 0.05), while UCi, UVi and VDF were positively correlated with MY (*p* < 0.0001; *r* = from 0.66 to 0.74). For the prediction of MY, the obtained equations had an *r^2^* ranging from 0.54 to 0.63. The UCi, UDf, UWi and UWf were included in these models (*p* < 0.05). It is concluded that there was an acceptable correlation (*r* = 0.60) between the measurements of the udder, the volume of the udder and the daily milk yield in Pelibuey sheep. When direct measurements of milk production cannot be performed in practice, the measurement of udders and their volume could be a viable alternative to estimate milk yield production as an indirect method.

## 1. Introduction

In the tropical regions of Latin America, sheep production systems are characterised by the use of animal genetic resources of native creole breeds, mainly hair sheep breeds [1]. In Mexico, the most common sheep breeds used in order of importance are the Pelibuey, Black Belly, Katahdin and Dorper breeds [1].

In the last decade, Pelibuey sheep have been the main maternal breed used for sheep breeding in the tropics of Mexico [2]. One of the first studies to determine the milk production and composition of milk from Pelibuey ewes was carried out by Castellanos and Valencia [3]. However, to date, few studies have evaluated these aspects in Pelibuey ewes and their crosses with Katahdin under tropical conditions in Mexico [4,5,6].

The quantification of milk production in sheep is important because milk is the main source of nutrients for the growth, development and health of lambs. If milk production is insufficient, the growth of lambs could be hampered [6,7,8]. For this reason, it is necessary to know the milk production in order to propose, if necessary, economically viable and profitable intervention strategies to increase milk production.

Many nutritional models use milk yield to determine the proper formulation of supplements and rations for ruminant animals based on their energy and nutrient needs [9,10]. However, in Pelibuey ewes, Ampueda and Combellas [11] reported that it was very difficult to milk the animals in order to measure the amount of milk produced, especially due to the small size of their teats. Although some studies have estimated milk production in hair breeds using direct and indirect methods, such as manual milking, mechanical milking and the double lamb weighing technique [5], it is important to evaluate other indirect methods that are easy to carry out. Among these indirect methods, udder measurements and the weight gain of lambs has been evaluated in addition to more sophisticated and expensive techniques, such as the dilution of isotopes [5,6,12,13,14]. The need to correctly assess milk yield is important, but there is a chronic lack of information [15,16] and a lack of dissemination of available and accurate methods [10].

The present study was based on udder measurements to determine udder volume by means of mathematical formulas, which could be a useful tool for predicting milk production. This method is viable, practical and low in cost, unlike many other techniques [6,12,17]. Additionally, good correlations were previously found between the size of teats and udders and the production of milk in different breeds of sheep [12,16,17,18]. Because no studies have used udder measurements to predict milk yield in hair sheep, the objective of the present study was to evaluate the relationship between udder measurements and milk production in Pelibuey ewes.

## 2. Material and Methods

### 2.1. Experimental Site, Animals and Collection of Udder Measurements 

The animals were treated in accordance with guidelines and regulations for animal experimentation of the División Académica de Ciencias Agropecuarias, Universidad Juárez Autónoma de Tabasco. The present study was approved by review committee approval number: PFI: UJAT-DACA-2015-IA-02.

The experiment was carried out at the Centro de Integración Ovina del Sureste (CIOS) located at 17° 78” N, 92° 96” W, 10 m above sea level and 29 km from the Villahermosa-Teapa highway in Tabasco, Mexico. Pelibuey ewes of differing body condition scores (BCS) on a scale of 1–5 were selected from a commercial farm, with a score of 1 indicating very thin and 5 obese [19]. Thirty-eight recently lambed (5–10 days), clinically healthy Pelibuey ewes (aged 2–3 years) with single (n = 33) and twin lambing (n = 5) were considered in the present study, with 36.34 kg ± 4.90 kg average body weight (BW) and an average BC of 1.5 ± 0.50. 

The ewes and their lambs were confined in raised slatted-floor cages in roofed buildings without walls. The food supply consisted of star grass hay (*Cynodon nlemfuensis*), ground corn, soybean meal, sugarcane molasses and minerals, with an estimated metabolisable energy of 12 MJ/kg DM and 15% crude protein [20]. Throughout the experiment the diet was offered ad libitum, with feeding levels designed to ensure a refusal margin of 10% each day. The diet was formulated to meet the requirements of dairy ewes with a mean BW of 45 kg and a mean milk yield of 1.74 kg/d. The crude protein (CP) and fat contents were 4.5% and 7.0%, respectively, according to the AFRC [20] equations. The objective was to maintain constant BWs and BCSs of ewes throughout the whole experimental period.

The daily milk yield (DMY, kg) of ewes was determined by hand milking from the second week after parturition until weaning of the lambs at 56 days. The lambs were separated daily from their dam at 19:00 h. During this period, the lambs had free access to feed (18% CP, 12 MJ ME/kg DM). After 12 h of separation, the ewes were milked after being injected IM with 3 IU of oxytocin. Before milking was performed, the teats of the animals were disinfected using an iodine solution and, after about 30 s, the teats were dried with paper towels. 

Before (i) and after (f) milking, the following udder measurements (cm) were recorded for each ewe as described by Emediato et al. [17] and Merkhan and Alkass [21]: Udder depth (UD), udder circumference (UC), udder width (UW), teat length (TL) and teat diameter (TD). For the measurements, a flexible fibreglass tape (Truper®, Truper S.A. de C.V., San Lorenzo, Mexico) and a digital calliper (Truper®, Truper S.A. de C.V., San Lorenzo, Mexico) were used. All measurements were taken twice a week (347 independent measurements). The udder volume (UV) was calculated according to the equation described by Izadifard and Zamiri [22]. Additionally, the difference (VDF) between initial UV (UVi) and the final UV (UVf) was calculated as VDF = UVi − UVf.
*R* = CP/2 × *π*(1)
UV = *π × R*^2^ × UD(2)
where *R* = the radius (cm), CP = the circumference perimeter (cm), *π* = 3.142, UV = udder volume (cm^3^) and UD = udder depth (cm).

### 2.2. Statistical Analyses

The descriptive statistical analysis was performed using the PROC MEANS procedure of SAS [23]. Correlation coefficients among variables were estimated using the PROC CORR procedure of SAS [23]. Regressions were developed using the PROC REG procedure of SAS [23]. The STEPWISE option and Mallow’s Cp were used in the SELECTION statement to select the variables included in the model. Data points were removed if their studentised residual was outside the range of −2.5 to 2.5. The effect of litter size was included in the statistical analyses as a covariate. 

According to Tedeschi [24], several statistics have been used to assess the predictability of the equations, including the coefficients of determination (r^2^), mean square error (MSE), standard deviation (SD), mean square error of prediction (MSEP) and root of the MSEP (RMSEP), in order to account for the difference between predicted values and true values. The mean bias (MB), as described by Cochran and Cox [25], was used as a representation of the average inaccuracy of the model. The modelling efficiency factor (MEF), which represents the proportion of variation explained by the line Y = X, was used as an indicator of goodness of fit [26,27]. The coefficient of model determination (CD) was used to assess variance in the predicted data. The bias correction factor (Cb), a component of the concordance correlation coefficient (CCC) [28], was used as an indicator of deviation from the identity line, and the CCCs were also used as a reproducibility index to account for accuracy and precision. High accuracy and precision were assumed when the coefficients were >0.80, and low accuracy and precision were assumed when the coefficients were <0.50. Finally, all calculations were obtained using the Model Evaluation System [24].

## 3. Results

Descriptive statistics for udder measurements and milk yield are presented in Table 1. A large variation in milk yield (0.100–1.04 kg/d) was observed, with a mean of 0.39 kg/d ± 0.18 kg/d. The UCi and the VDF also widely ranged from 25.80 cm to 53.30 cm and from 1.030 cm^3^ to 2418 cm^3^, respectively. 

The UWf, TDi, TDf, TLi and TLf were not correlated with milk yield (*p* > 0.05). A moderate correlation was found between milk yield and UCi (*r* = 0.66) and between milk yield and the UDi (*r* = 0.47) and UWi (*r* = 0.49) (Table 2). A positive correlation (*p* < 0.0001) was found between milk yield with UVi (*r* = 0.71) and VDF (*r* = 0.74). For the prediction of milk yield, the obtained equations had an *r^2^* that ranged from 0.54 to 0.63 (Table 3). The UCi, UDf, UWi and UWf were included in these models (*p* < 0.05).

The predictive equations obtained in the current study were moderately precise (*r^2^* = 0.55 to 0.63) but highly accurate (Cb = 0.85 to 0.97), with a reproducibility index ranging from 0.68 to 0.78 (Table 4). The MEF indicated a moderate efficiency of prediction (from 0.47 to 0.63). The CD ranged from 1.56 to 1.89, indicating high variability in the predicted data (Table 4), whereas the partition of the % MSEP indicated that the mean bias fluctuated from 0.001 to 28.50 and that the systematic bias varied from 0.01% to 1.58% (Table 4).

## 4. Discussion

In the present study, the viability of using udder measurements to predict milk production in Pelibuey ewes was evaluated. The literature contains few studies reporting milk production in hair sheep breeds, such as the Pelibuey and their crosses [4,5,6], in which milk yields around 1.1–1.74 kg/d have been reported. Although some information is available on milk yield in this breed, no reliable non-nvasive tools have been validated to predict milk yield in this breed. The udder measurements have been used to estimate milk production in dairy ewes of several sheep breeds [12,17,18,28,29]. However, this is the first study that has reported the use of udder measurements to predict milk yield in Pelibuey ewes. 

Emediato et al. [17] reported positive correlations in Bergamasca ewes between daily milk yield and udder depth, circumference and width (*r* = 0.74, 0.75 and 0.62, respectively). Ayadi et al. [18] evaluated mammary morphology and milk production in Sicilia-Sarde dairy sheep. Daily milk yield was positively and significantly correlated with UV and UD. Similar to our results, these authors did not observe significant correlations among daily milk yield and diameter or length of teats. Likewise, Sezenler et al. [30] found positive and significant correlations between DMY and udder circumference and udder width, indicating that udder measurements could serve as a predictor of milk production. Likewise, the evaluations carried out by Angeles et al. [29] reported an association between different morphological traits of body and udders of Spanish Assaf sheep. In particular, udder traits such as depth (*r^2^* = 0.47) and length (*r^2^* = 0.25) were the most correlated with milk production. These studies are in agreement with the findings of Iñiguez et al. [12] for Awassi sheep and their crosses, where total milk production and total protein, fat and nonfat solids were positively correlated (*r* = 0.36 to 0.76) with udder circumference and udder width, which is similar to our results, where we found a positive moderate correlation among UWi, UCi and MY (*r* = 0.49 to 0.66, respectively). 

Positive relationships between udder measurements and milk production have been reported in other species. In dromedary camels, Musaad et al. [31] reported that udder depth, udder circumference and distance between teats were positively correlated with milk production and were significantly affected (*p* ≤ 0.05) by the lactation stage. Merkhan and Alkass [21] evaluated Black and Meriz goats and found that milk production was correlated with udder circumference and udder length (*p* < 0.01) in both breeds. In Saanen goats, Linzell [32] reported a high correlation (*r* = 0.87) between udder volume and milk production. Capote et al. [33] also evaluated the correlations between udder morphology and milk yield in Tinerfen dairy goats and obtained a positive correlation (*r* = 0.79) between udder volume and milk production. On the contrary, in Sicilia-Sarden sheep, Ayadi et al. [18] did not find any relationship between the distance between teats (*p* > 0.05) and milk production.

Udder morphology measurements, their relationships with milk production traits and their usefulness for genetic improvement has been studied in different sheep breeds [12,16]. Among these measurements, udder circumference and teat width has been shown to be significantly correlated with total milk yield and are considered good predictors of performance in several sheep breeds [12]. In addition, Capote et al. [34] indicated a greater importance on the balance between the horizontal and vertical diameters of udder compared with length parameters. Other authors [12,16,18,32,33,34] have suggested that the predictive capacity of udder measurements have important practical implications for breeding programmes to increase resilience to mastitis in the population [35]. In addition, Milerski et al. [36] reported that the linear scores for udder depth, cistern depth, teat position and teat size would permit prediction of future correlated responses in milk-oriented selection schemes in the Tsigai, Improved Walachian and Lacaune breeds. Ewes with ideal udder sizes (MY = 1.04, UCi = 53.30, UDf = 20, UWi = 15.5, UWf = 15.2, VDF = 2418) could be selected to improve milk production. Our analysis was highly accurate (Cb > 0.852 and RMSEP > 29%) in predicting milk yield according to the selected udder measurements, but the precision (r^2^ < 0.632) was systematically low, resulting in under prediction (CD < 1.897) across equations. The variation in milk yield may be largely explained by environmental effects [6] or genotype [6,15,37]. Although the previous studies indicated that ewes rearing twin lambs had higher daily milk yields [6,38], because of low prolificacy in the present study (1.13), the effect of litter size could not be determined due to insufficient statistical power. Likewise, the animals used in this study were very thin, 1.5 ± 0.50 points of BCS (on a scale of 1 to 5). However, this BCS is characteristic of tropical sheep-production systems based on grazing in the dry season [1]. Although we obtained a moderate and positive correlations estimates among udder measurements and milk yield in Pelibuey ewes in the current study, more work is required to explore the other factors that affect milk production (level of feeding, animal health, litter size, BCS, body mass index, genetics) in hair sheep ewes raised in tropical production systems. Therefore, for predictive purposes, these factors could also be considered to achieve precise and accurate prediction of milk yield.

## 5. Conclusions

The present study obtained moderate and positive correlations estimates among udder measurements, udder volume and daily milk yield in Pelibuey ewes. The udder measurements and its volume could be a useful tool to predict milk yield.

## Figures and Tables

**Table 1 animals-10-00518-t001:** Descriptive statistics for udder measurements and milk yield in Pelibuey ewes.

Variable	Description	Mean ± SD	Maximum	Minimum
MY	Daily milk yield (kg)	0.39 ± 0.18	1.04	0.10
UCi	Initial udder circumference (cm)	39.61 ± 4.99	53.30	25.80
UCf	Final udder circumference (cm)	33.38 ± 4.59	45.0	24.30
UDi	Initial udder height (cm)	13.91 ± 2.51	20.30	5.00
UDf	Final udder height (cm)	12.51 ± 2.23	20.0	5.70
UWi	Initial udder width (cm)	11.68 ± 1.55	15.5	7.10
UWf	Final udder width (cm)	9.53 ± 1.79	15.2	1.30
TDi	Initial diameter of the teat (cm)	1.58 ± 0.62	2.90	1.00
TDf	Final diameter of the teat (cm)	1.49 ± 0.51	2.70	1.00
TLi	Initial length of the teat (cm)	2.64 ± 1.41	4.50	1.20
TLf	Final length of the teat (cm)	2.51 ± 0.37	3.70	1.20
UVi	Initial udder volume (cm^3^)	1790 ± 652.58	4271	664.47
UVf	Final udder volume (cm^3^)	1137 ± 384.81	2862	458.84
VDF	Difference between UV (UVi − UVf, cm^3^)	652.42 ± 446.88	2418	1.03

The suffix initial and final indicate that the measures were taken before (i) and after (f) every milking.

**Table 2 animals-10-00518-t002:** Correlation coefficients among measured variables for udder measurements and mean daily milk yield in Pelibuey ewes.

	UCf	UDi	UDf	UWi	UWf	TDi	TDf	TLi	TLf	UVi	UVf	VDF	MY
UCi	0.68 ***	0.31 ***	0.24 *	0.61 ***	0.37 ***	−0.05^ns^	−0.06^ns^	0.05^ns^	0.06^ns^	0.85 ***	0.69 ***	0.66 ***	0.66 ***
UCf	1.00	0.18^ns^	0.12^ns^	0.38 ***	0.37 ***	−0.06 ***	0.02^ns^	−0.02^ns^	0.04^ns^	0.56 ***	0.79 ***	0.14 ***	0.24 ***
UDi		1.00	0.63 ***	0.16^ns^	0.06^ns^	−0.030^ns^	0.003^ns^	0.08^ns^	0.14^ns^	0.72 ***	0.48 ***	0.65 ***	0.47 ***
IDf			1.00	0.15^ns^	0.07^ns^	−0.033^ns^	−0.03^ns^	0.06^ns^	0.14^ns^	0.49 ***	0.65 ***	0.16^ns^	0.27 ***
UWi				1.00	0.46 ***	0.07^ns^	0.04^ns^	0.08^ns^	0.016^ns^	0.51 ***	0.41 ***	0.39 ***	0.49 ***
UWf					1.00	0.005^ns^	0.05^ns^	0.01^ns^	0.05^ns^	0.31 ***	0.37 ***	0.13^ns^	0.15^ns^
TDi						1.00	0.08^ns^	0.07^ns^	0.18*	−0.03^ns^	−0.05^ns^	−0.007^ns^	0.003^ns^
TDf							1.00	0.01^ns^	0.24 ***	−0.03^ns^	0.006^ns^	−0.05^ns^	−0.05^ns^
TLi								1.00	0.19^ns^	0.07^ns^	0.009^ns^	0.09^ns^	0.12^ns^
TLf									1.00	0.11^ns^	0.10^ns^	0.07^ns^	0.11^ns^
UVi										1.00	0.74 ***	0.81 ***	0.71 ***
UVf											1.00	0.22 ***	0.35 ***
VDF												1.00	0.74 ***

* *p* < 0.05; *** *p* < 0.0001; NS: Not significant, *p* > 0.05; MY: Daily milk yield; UCi: Initial udder circumference; UCf: Final udder circumference; Udi: Initial udder height; UDf: Final udder height; UWi: Initial udder width; UWf: Final udder width; TDi: Initial diameter of the teat; TDf: Final diameter of the teat; TLi: Initial length of the teat; TLf: Final length of the teat; UVi: Initial udder volume; UV: Final udder volume; VDF: Difference between UV (UVi − UVf, cm^3^).

**Table 3 animals-10-00518-t003:** Regression equations for predicting milk yield (MY, kg/d) according to udder measurements in Pelibuey ewes.

#	Equation	n	MSE	RMSE	*r^2^*	*p*
3	MY (kg) = 0.194 (± 0.012***) + 0.00031 (± 0.000015 ***) × UDf	347	0.016	0.126	0.54	0.0001
4	MY (kg) = −0.204 (± 0.065^*^) + 0.011 (± 0.001***) × UCi + 0.0002 (± 0.00001***) × VDF	347	0.014	0.118	0.60	0.0001
5	MY (kg) = −0.276 (± 0.063***) + 0.008 (± 0.001***) × UCi + 0.017 (± 0.005^*^) × UWi + 0.0002 (± 0.00001***) × VDF	347	0.013	0.114	0.61	0.0001
6	MY (kg) = −0.356 (± 0.066***) + 0.006 (± 0.001*) × UCi + 0.009 (± 0.002*) × UDf + 0.017 (± 0.005*) × UWi + 0.0002 (± 0.00001***) × VDF	347	0.013	0.114	0.62	0.0001
7	MY (kg) = −0.34 (± 0.66***) + 0.007 (± 0.001***) × UCi + 0.009 (± 0.002*) × UDf + 0.02 (± 0.005***) × UWi − 0.009 (± 0.004*) × UWf + 0.0002 (± 0.00001***) × VDF	347	0.013	0.114	0.63	0.0001

**p* < 0.05; ****p* < 0.0001; MSE: Mean square error; RMSE: Root mean square error; *r^2^*: Coefficient of determination; MY: Daily milk yield; UCi: Initial udder circumference; UDf: Final udder height; UWi: Initial udder width; UWf: Final udder width; VDF: Difference between UV (UVi − UVf, cm^3^).

**Table 4 animals-10-00518-t004:** Mean and descriptive statistics of the accuracy and precision of the equations for predicting milk yield based on udder measurements and udder volume versus the observed milk yield in Pelibuey ewes.

Variable ^1^	Obs	(Equation (3))	(Equation (4))	(Equation (5))	(Equation (6))	(Equation (7))
Mean	0.396	0.396	0.362	0.369	0.323	0.389
SD	0.188	0.138	0.132	0.134	0.132	0.150
Maximum	1.040	0.944	0.850	0.844	0.790	0.929
Minimum	0.095	0.194	0.090	0.068	0.036	0.047
*r^2^*	---	0.55	0.59	0.61	0.62	0.63
CCC	---	0.71	0.71	0.73	0.68	0.78
Cb	---	0.95	0.92	0.93	0.85	0.97
MEF		0.55	0.56	0.58	0.47	0.63
CD		1.86	1.89	1.88	1.56	1.56
Regression analysis						
Intercept (β_0_)						
Estimate	---	−0.003	−0.002	−0.009	0.031	0.009
SE	---	0.020	0.018	0.018	0.016	0.017
*p* value (β_0_ = 0)	---	0.88	0.90	0.63	0.06	0.56
Slope (β_1_)						
Estimate	---	1.01	1.10	1.09	1.13	0.99
SE	---	0.049	0.048	0.046	0.047	0.040
*p* value (β_1_ = 1)	---	0.85	0.04	0.04	0.01	0.89
MSEP source, % MSEP						
Mean bias	---	0.001	7.58	4.96	28.51	0.45
Systematic bias	---	0.010	1.15	1.16	1.58	0.01
Random error	---	99.98	91.27	93.89	69.91	99.54
Root MSEP						
Estimate	---	0.13	0.12	0.12	0.14	0.11
% of the mean	---	31.93	34.47	32.66	42.57	29.42

^1^ Obs: Observed evaluation data set; SD: standard deviation; CCC: Concordance correlation coefficient; Cb: Bias correction factor; MEF: Modelling efficiency; CD: Coefficient of model determination; MSEP: Mean square error of the prediction; (Equations (3)–(7)): in the Table 3.

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
