# Peer review of "Udder Measurements and Their Relationship with Milk Yield in Pelibuey Ewes"

_animals, 2020, doi:10.3390/ani10030518_

Round 1

Reviewer 1 Report

The manuscipt describe a relationship between udder measurements and milk yield in Pelibuey breed sheep. As the authors well described in the discussion, this kind of research is not original because there are a lot of work that evaluate in Bergamasca or Sicilia sarda dairy sheep the correllation between the udder measurements and milk yeld. For this reason I suggeste to the authors to improve the results section with some result about the quality of milk to give originality to the manuscript.

Author Response

Reviewer #1:The manuscipt describe a relationship between udder measurements and milk yield in Pelibuey breed sheep. As the authors well described in the discussion, this kind of research is not original because there are a lot of work that evaluate in Bergamasca or Sicilia sarda dairy sheep the correllation between the udder measurements and milk yeld. For this reason I suggeste to the authors to improve the results section with some result about the quality of milk to give originality to the manuscript.

We declarate that this is the first study that reports the use of udder measurements to predict milk yield in Pelibuey ewes.

Reviewer 2 Report

The study is aimed at evaluation of usefulness of udder measurements for indirect selection for milk production in Pelibuey sheep. Moderate correlations between udder size traits and milk yield were obtained.

Broad comments

The work is interesting and well written. Nevertheless, for further work on this topic I would recommend authors to think about modification of the formula for the udder volume:

UV=π *R2*UD

In fact it is a formula for cylinder volume, but the sheep udder have more semispheral shape. Additionally in selection on udder size is necessary to focus more on udder circumference or udder width than on udder depth  in course do not obtain baggy udders, which are problematic both for milking and for lamb suckling. In used formula (practically selection index) UD plays too important role.

In the work the phenotypic correlations are computed practically on row data. It would be interesting to compute partial correlations after adjusting the data according the systematic factors e.g. parity or days in milk.

From a formal point of view, I recommend adjusting the abbreviations for udder measurements to make them clearer. All measurements were done twice: before (initial) and after milking (final).  That is why for me it would be better to distinguish them for example  by index: UDi, UDf or UCi, UCf not to change completely this abbreviations (IC, FC – for udder circumference initial and final).

I recommend also to use the same number of decimal places for a particular character. For example, in the VDF range in the abstract varied from 1.030 to 2418 - I am not sure if there is not a typo error so I recommend putting 1 to 2418. The same problem applies to Tables 1 and 4.

Specific comments

L32 – „strategies of for the lambs“ – I would propose „for“ instead of „of“.

L39- I propose „…varied from 1 to 2418 cm3“– see broad comments.

L43- (r=60) – I would advise to use more concrete figures in this place; e.g. the range of correlations or their highest value.

L65 – …it is very difficult to milk…

L88 – aabove

L230 – measurements

Table 2 - it does not appear entirely, please adjust to the width of the page.

Table 3 – r2 for equations 1 and 4 are not rounded correctly; according figures in the Table 4 they have to be 0.55 and 0.63 or use 3 decimals as in Table 4.

Author Response

The study is aimed at evaluation of usefulness of udder measurements for indirect selection for milk production in Pelibuey sheep. Moderate correlations between udder size traits and milk yield were obtained.

Broad comments

The work is interesting and well written. Nevertheless, for further work on this topic I would recommend authors to think about modification of the formula for the udder volume:

UV=π *R2*UD

In fact it is a formula for cylinder volume, but the sheep udder have more semispheral shape. Additionally in selection on udder size is necessary to focus more on udder circumference or udder width than on udder depth in course do not obtain baggy udders, which are problematic both for milking and for lamb suckling. In used formula (practically selection index) UD plays too important role.

AU: This formula has been  used in other studies relate with udder volume estimation in ewes.

In the work the phenotypic correlations are computed practically on row data. It would be interesting to compute partial correlations after adjusting the data according the systematic factors e.g. parity or days in milk.

AU: this is the first study that reports the use of udder measurements to predict milk yield in Pelibuey ewes. In others studies cuold be evaluate the factors describe above.

From a formal point of view, I recommend adjusting the abbreviations for udder measurements to make them clearer. All measurements were done twice: before (initial) and after milking (final).  That is why for me it would be better to distinguish them for example  by index: UDi, UDf or UCi, UCf not to change completely this abbreviations (IC, FC – for udder circumference initial and final).

R: This has been corrected according to the reviewer’s comments.

I recommend also to use the same number of decimal places for a particular character. For example, in the VDF range in the abstract varied from 1.030 to 2418 - I am not sure if there is not a typo error so I recommend putting 1 to 2418. The same problem applies to Tables 1 and 4.

R: This has been corrected according to the reviewer’s comments.

Specific comments

L32 – „strategies of for the lambs“ – I would propose „for“ instead of „of“.

R: This has been corrected according to the reviewer’s comments.

L39- I propose „…varied from 1 to 2418 cm3“– see broad comments.

R: This has been corrected according to the reviewer’s comments.

L43- (r=60) – I would advise to use more concrete figures in this place; e.g. the range of correlations or their highest value.

R: This has been corrected according to the reviewer’s comments.

L65 – …it is very difficult to milk…

R: This has been corrected according to the reviewer’s comments.

L88 – aabove

R: This has been corrected according to the reviewer’s comments.

L230 – measurements

R: This has been corrected according to the reviewer’s comments.

Table 2 - it does not appear entirely, please adjust to the width of the page.

R: This has been corrected according to the reviewer’s comments.

Table 3 – r2 for equations 1 and 4 are not rounded correctly; according figures in the Table 4 they have to be 0.55 and 0.63 or use 3 decimals as in Table 4.

R: This has been corrected according to the reviewer’s comments.